# The Efficacy of Three Types of Disinfectants on the Microbial Flora from the Surface of Impression Materials Used in Dentistry—In Vitro Study

Diana Cerghizan [1][iD], Kinga Mária Jánosi [1,*][iD], Cristina Nicoleta Ciurea [2][iD], Oana Popelea [1], Monica Dora Baloş [1], Adriana Elena Crăciun [1], Liana Georgiana Hănţoiu [1] and Aurița Ioana Albu [1]

1   Faculty of Dental Medicine, George Emil Palade University of Medicine, Pharmacy, Science, and Technology of Targu Mures, 38 Gh. Marinescu Str., 540142 Targu Mures, Romania
2   Department of Microbiology, George Emil Palade University of Medicine, Pharmacy, Science, and Technology of Targu Mures, 38 Gh. Marinescu Str., 540142 Targu Mures, Romania
*   Correspondence: kinga.janosi@umfst.ro; Tel.: +40-740165717

**Abstract:** During impressions, bacteria, viruses, and fungi remain on the impression material, representing a significant risk for the medical team (dentists, dental assistants, and laboratory technicians). Impression disinfectants have been introduced into dentistry to reduce the risk of cross-infection. This study was performed by examining the surface disinfection of five commonly used impression materials in prosthodontics: alginate-Tropicalgin (Zhermack®), condensation silicone-Zetaplus (Zhermack®), Oranwash L (Zhermack®); and addition silicone-Elite HD + Putty Soft (Zhermack®), Elite + Light Body (Zhermack®) after the disinfection with three disinfectants: Zeta 3 Soft (Zhermack®), Zeta 7 Spray (Zhermack®) and Zeta 7 Solution (Zhermack®). Before disinfection, the impression materials were contaminated with *Staphylococcus aureus* ATCC 25923, *Pseudomonas aeruginosa* ATCC 27853, and *Candida albicans* ATCC 10213. Our results demonstrate the efficacy of the two examined specific disinfectants in reducing cross-infection risk. The surface disinfectant-spray is not adequate for impression disinfection. A high number of bacterial colonies were found on the surface of all impressions disinfected with this spray. The immersion-disinfection method effectively eliminates the risk of cross-infections (Kruskal–Wallis test showed a $p < 0.001$).

**Keywords:** infection prevention; cross-infection; disinfectants; dental impression

## 1. Introduction

Prosthetic rehabilitation is a sequence of clinical-technical procedures through which the operator comes into contact with various fluids from the oral cavity. This is through instruments, materials, and aerosols generated during tooth preparation.

Based on scientific research, more than 700 species of bacteria, fungi, viruses, and protozoa are potentially to be found in the oral cavity [1], from which only 54% have been cultivable and identified [2]. According to Chidambaranathan AS et al. (2017), 67% of dental impressions are contaminated with *Streptococcus* spp., *Staphylococcus* spp., *Candida* spp., methicillin-resistant *Staphylococcus aureus* (MRSA), or *Pseudomonas aeruginosa* [3].

During impressions, bacteria, viruses, and fungi remain retained on the impression material, representing a significant risk for the medical team (dentists, dental assistants, and laboratory technicians) [4]. Disinfection means the complete elimination of vegetative forms of microorganisms, except for the bacterial spores from inanimate objects. Sterilization is the complete elimination of all microorganisms and spores [5]. Disinfectants can be low-level (phenolics, quaternary ammonium compounds, diluted glutaraldehyde, 3% hydrogen peroxide), intermediate-level (alcohol, iodine, iodophor), and high-level (400–450 ppm hypochlorous acid, >2% glutaraldehyde, 7.5% hydrogen peroxide, 650–675 ppm hypochlorite) [5]. Impression disinfectants have been introduced in dentistry

to reduce the risk of cross-infection. Disinfection of impressions, before transport to the dental laboratory, is essential [6]. Sometimes the disinfection of the impressions is carried out with surface disinfectants, which can cause surface modifications and dimensional changes, resulting in inadequate restorations [7,8]. The British Dental Association recommends rinsing the impressions with water after the removal from the oral cavity. A large percentage of microorganisms can persist on the surfaces of the materials. This method is not adequate for preventing cross-infections [9,10]. An adequate disinfection method must consider the solution type, concentration, and action time [11].

The aim of the study is to compare three commonly used disinfectants and their antimicrobial effect on impression-material surfaces.

## 2. Materials and Methods

This study was performed by examining the surface disinfection efficacy of five dental impression materials: alginate-Tropicalgin (Zhermack®, Badia Polesine, Italy); condensation silicone-Zetaplus (Zhermack®) and Oranwash L (Zhermack®); and, in addition, silicone-Elite HD + Putty Soft (Zhermack®), and Elite HD + Light Body (Zhermack®) (Table 1), after the disinfection with three disinfectant solutions used for the infection control.

**Table 1.** The used-impression-material characteristics, according to the manufacturer (Zhermack®).

| Impression Materials | Type | Consistency | Delivery System | Clinical Working Time (min:s) | Time in the Mouth (min:s) | Setting Time (min:s) |
|---|---|---|---|---|---|---|
| Tropicalgin | Alginate | - | Manual mixing | 1:35 | 1:00 | 2:35 |
| Zetaplus | C-silicone | Putty | Manual mixing | 1:15 | 3:30 | 4:45 |
| Oranwash L | C-silicone | Light-body | Manual mixing | 1:30 | 3:30 | 5:00 |
| Elite HD+ Putty Soft | A-silicone | Putty | Manual mixing | 2:00 | 3:30 | 5:30 |
| Elite HD+ Light Body | A-silicone | Light-body | Dispenser gun | 2:00 | 3:30 | 5:30 |

The manufacturer defines the illustrated setting time for the mix, starting at 23 °C. In this in vitro study, all the impression materials were prepared at room temperature. A higher or lower temperature or the inaccurate dosage of the catalyst can modify this setting intervallum. High temperatures or overuse of catalysts can speed up hardening, low temperatures or insufficient use of catalysts can slow it down.

A surface disinfectant and two specific disinfectants were used for the disinfection of the impressions: Zeta 3 Soft (Zhermack®); Zeta 7 Spray (Zhermack®); and Zeta 7 Solution (Zhermack®) at 1:100 dilution (Table 2). Zeta 3 Soft is a disinfectant for instruments and surfaces sometimes used incorrectly to disinfect impressions.

A plastic plug, disinfected with Zeta 3 Soft for 5 min, was used to obtain the specimens from the impression materials. The specimens were washed with water to remove the residual disinfectant that might interfere with the impression materials. The impression materials were handled using nitrile gloves (the hardening of A-silicones is inhibited through contact with latex gloves), and were processed according to the manufacturer's recommendations: Zetaplus and Elite HD + Putty Soft were manually mixed, Oranwash L was prepared with a spatula on new mixing pads, Elite HD+ Light Body was prepared from an automix cartridge. Tropicalgin was prepared by using a disinfected plastic bowl and mixing spatula.

**Table 2.** The used-disinfectant properties, according to the manufacturer (Zhermack®).

| Disinfectants | Application | Active Ingredients/100 g | Action Time (Minutes) | Spectrum of Action |
|---|---|---|---|---|
| Zeta 3 Soft | Ready-to-use spray | Alcohols (34.4 g ethanol, 14 g isopropanol) | 1–5 | Bactericidal: EN 13727, EN 14561 (*Staphylococcus aureus, Pseudomonas aeruginosa*) Yeasticidal: EN 13624, EN 14562 (*Candida albicans*) Tuberculocidal: EN 14348, EN 14563 Virucidal: EN 14476 |
| Zeta 7 Spray | Ready-to-use spray | Alcohols (83 g ethanol, 10 g 2-propanol) | 3 | Bactericidal: EN13727 (*Staphylococcus aureus, Pseudomonas aeruginosa*) Yeasticidal: EN 13624 (*Candida albicans*) Tuberculocidal: EN 14348, EN 14563 Virucidal: EN 14476 |
| Zeta 7 Solution | Concentrated solution (recommended dilution 1%) | Quaternary ammonium salts, Phenoxyethanol (7.7 g dimethyl-didecyl-ammonium chloride, 15 g phenoxyethanol) | 10 | Bactericidal: EN13727 (*Staphylococcus aureus, Pseudomonas aeruginosa*) Yeasticidal: EN 13624 (*Candida albicans*) Tuberculocidal: EN 14348 Virucidal: EN 14476 |

In the case of the Zetaplus, the measuring spoon was used to dose the required quantity of material, which was spread out over the operator's hand, folded and kneaded energetically with the fingertips for approximately 30 s until the color was even, without any stripes. For each measure of Zetaplus material, two strips of Zhermack Indurent Gel were added along the measuring scoop (4 cm). In the case of the Elite HD + Putty Soft, the two components were taken in equal proportions of base and catalyst from the cans, using only the respective measuring spoons. The materials were mixed similarly to the Zetaplus until a mass of uniform color without streaks was obtained. For the dosage of the Oranwash L, the dosage scale was used on the mixing block, and an equal quantity of Zhermack Indurent Gel was added to the length of the fluid material. The two components were mixed for 30 s with a spatula by pressing the mixture over the mixing block to eliminate any air which may have been incorporated into the mixture. The procedure was repeated until the mix had a homogeneous color.

The Elite HD + Light Body cartridge was mounted on a dispensing gun. Before assembling the mixing tips, it was verified that the two components (base and catalyst) would come out uniformly, by exerting a slight pressure on the dispenser lever and extruding a small amount of material, which was removed. Then the mixing tip was inserted into the cartridge, and the material was extruded.

Tropicalgin was extracted from the package with the help of the measuring spoon after shaking the bag. For each spoon full of powder placed in the bowl, a 1/3 measure of water was added, then mixed with a plastic spatula until the color and consistency were homogenous.

The plug was fully filled with the prepared materials, and a pressure similar to the intraoral conditions was applied on the impression material. After the recommended setting time, the impression materials were checked clinically, to see if they were properly set, as is done in our daily practice. After retrieving the complete set of specimens, they were removed and placed in disinfected trays.

Thirty specimens of each impression material were obtained, totalizing 150 impressions. The specimens were 29 mm in diameter and 15 mm in height. The height of the specimens simulates the maximum height of the impression material in the impression tray at the margins and the edentulous spaces.

The specimens from each impression material were divided into three groups according to the disinfectants used, and placed in three separate trays: a. Zeta 3 Soft, b. Zeta 7 Spray, and c. Zeta 7 Solution. Each tray contained ten specimens from each impression material.

The following microorganisms were used to contaminate the impressions: *Staphylococcus aureus* ATCC 25923, *Pseudomonas aeruginosa* ATCC 27853, and *Candida albicans* ATCC 10213 (Table 3).

**Table 3.** Basic characteristics of the microorganisms.

| Microorganisms | Type | Shape | Frequently Found |
|---|---|---|---|
| *Staphylococcus aureus* | Gram-positive bacteria | Spherical-shaped | Upper respiratory tract, skin |
| *Pseudomonas aeruginosa* | Gram-negative bacteria | Rod-shaped | Soil, water, skin |
| *Candida albicans* | Fungi | | Gastrointestinal tract, mouth |

All these strains were obtained from the Culture Collection of the Department of Microbiology, George Emil Palade University of Medicine, Pharmacy, Science, and Technology of Târgu Mureș, Romania.

The disk-diffusion method was used to determine the microorganism's susceptibility to the used disinfectants, according to the CLSI (Clinical and Laboratory Standards Institute) and EUCAST (European Committee for Antimicrobial Susceptibility Testing) guidelines and the manufacturer's indications. This qualitative method is among the most flexible susceptibility-testing methods for antimicrobial agents. The method consists of placing paper disks saturated with antimicrobial agents on a lawn of bacteria, seeded on the surface of an agar medium, incubating the plate overnight, and measuring the presence or absence of a zone of inhibition around the disks [12]. In accordance with this procedure, *Staphyloccocus aureus*, *Pseudomonas aeruginosa,* and *Candida albicans* were inoculated separately on Mueller–Hinton culture media [13]. The laboratory procedure was performed according to the Kirby–Bauer disk-diffusion-susceptibility-test protocol of the American Society for Microbiology. The culture media were at room temperature prior to inoculation. Sterile rayon swabs were dipped in the microbial suspension. In the case of gram-negative bacteria, the excess fluid was removed by pressing and turning the swab against the inside of the tube to avoid over-inoculation. The inoculum was evenly spread over the entire agar surface, without gaps between streaks. Three sterile paper discs containing 7 μL of each disinfectant were placed on each culture-medium surface within 15 min of inoculation. Disks were placed in close and firm contact with the agar surface, and were not moved once applied. The plates were inverted to confirm that the disks would not fall off the agar surface. The 24 h incubation at 35 °C had to begin within 15 min of the disk application. The results were read after 24 h. For basic research, a mix of these microorganisms was used, after detecting the susceptibility of each microorganism to the studied disinfectants. The colonies were inoculated in saline solution and standardized using the McFarland scale (0.5 McFarland = $1–2 \times 10^8$ CFU/mL) [14]. The American Society of Microbiology protocol for obtaining the 0.5 McFarland standard consists of verifying the correct density of the turbidity standard by measuring absorbance, using a spectrophotometer with a 1 cm light path and matching cuvette. The absorbance at 625 nm must be 0.08 to 0.13 for the 0.5 McFarland standard. An equimolecular mixture was made with the three microorganisms (Figure 1)

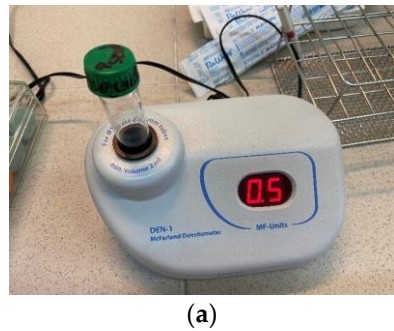

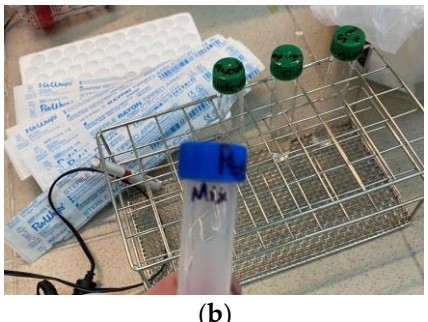

(**a**) (**b**)

**Figure 1.** Preparation of the microbial suspensions: (**a**) obtaining the 0.5 McFarland standard for each microorganism; (**b**) the equimolecular microbial-mix obtained from the three 0.5 McFarland suspensions.

A total of 150 sterile containers of 60 mL (Nantong Bestreatm Medical Instrument Co. Ltd., Nantong, China) were prepared. Each contained 20 mL of saline solution, 20 μL of microorganism suspension, and one impression-material specimen (Figure 2).

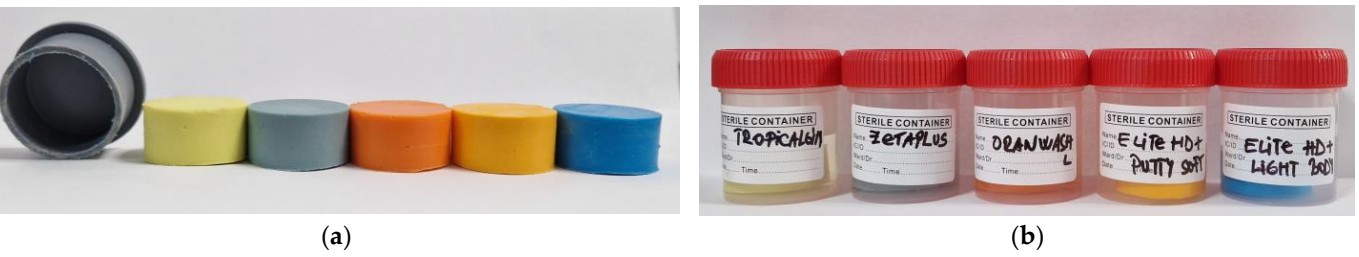

(**a**) (**b**)

**Figure 2.** Preparation of the specimens: (**a**) the plastic plug and one specimen from each impression material; (**b**) the containers with the microorganisms and the specimens.

Each container was shaken well for 10 s. The impression-material specimens were removed from the microorganism suspension after 10 min. This is considered the maximum handling time in a dental office, from the beginning of the impression procedure until the final impression would be disinfected. After rinsing the specimens with water for 30 s, they were organized into three different trays. Each tray contained ten specimens of each of the five impression materials. Control samples were collected from each specimen using sterile rayon swabs, according to the recommendation of the American Society of Microbiology, pressed onto the surface, and moved in two directions (right and left) at right angles. The stick of the swab was rotated between the thumb and forefinger for 15 s. The disinfectants were applied from 15–20 cm distance until the whole surface of the specimens was equally wet, and left to act according to the manufacturer's recommendations: 5 min after spraying for Zeta 3 Soft, 3 min after spraying for Zeta 7 Spray, and 10 min of immersion for Zeta 7 Solution. After this period, the impressions were rinsed for 10 s with water. Sterile rayon swabs were used to collect the samples from the disinfected impression-materials surface, using the same swabbing technique as for the control samples.

The Mueller–Hinton medium was used for the inoculation, which was performed with the same technique as in the case of the disk-diffusion method. All culture media were incubated at 35 °C. The results were read after 24 h, by counting the number of the growing colonies (from the control samples and the samples obtained after disinfection) traditionally, by using a pen and a click counter. The working protocol was repeated ten times for each impression material and disinfectant. The obtained values were recorded and compared statistically, using the GraphPad Prism 9 for macOS version 9.3.1 (350) (San Diego, CA, USA). The Kruskal–Wallis test, followed by Dunn's multiple-comparisons test, was used for statistical evaluation. The statistical significance was set at $p < 0.05$.

### 3. Results

The disc-diffusion method showed that all microorganisms are sensitive to the disinfectants involved in the study.

All control samples presented microbial load after 10 min of immersion in the microbial suspension and rinsing with water for 30 s (Table 4).

**Table 4.** The mean values of microbial colonies in the case of the control group for each impression material at 1:10 dilution.

| | Tropicalgin | Zetaplus | Oranwash L | Elite HD + Putty Soft | Elite HD + Light Body |
|---|---|---|---|---|---|
| Minimum | 161.0 | 180.0 | 191.0 | 200.0 | 20.0 |
| Median | 215.0 | 214.0 | 192.0 | 212.0 | 206.0 |
| Maximum | 218.0 | 225.0 | 203.0 | 212.0 | 225.0 |
| Mean | 198.0 | 206.3 | 195.3 | 208.0 | 210.7 |
| Std. Deviation | 32.08 | 23.46 | 6.658 | 6.928 | 12.66 |
| Lower 95% CI of mean | 118.3 | 148.1 | 178.8 | 190.8 | 179.2 |
| Upper 95% CI of mean | 277.7 | 264.6 | 211.9 | 225.2 | 242.1 |

In the case of the control groups, by applying the one-way ANOVA (Kruskal–Wallis test) with the post hoc Dunn's test, there was no statistical difference between the mean values of the number of colonies.

A reduced number of microbial colonies were grown on the culture media in the case of the two specific disinfectants (Zeta 7 Solution and Zeta 7 Spray) (Figures 3 and 4), and only in the case of the surface disinfectant were a higher number of colonies obtained (Figure 5).

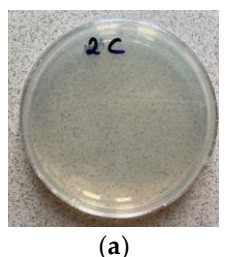 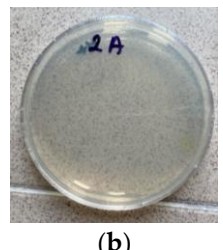 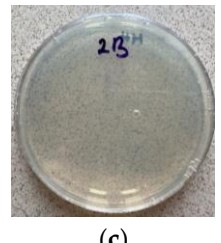 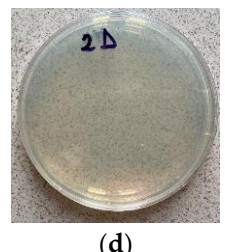 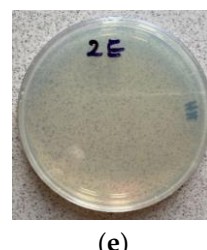

(a)     (b)     (c)     (d)     (e)

**Figure 3.** Microbial colonies after the disinfection with Zeta 7 Solution for each impression material: (**a**) Tropicalgin; (**b**) Zetaplus; (**c**) Oranwash L; (**d**) Elite HD + Putty Soft; (**e**) Elite HD + Light Body.

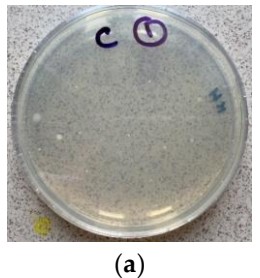 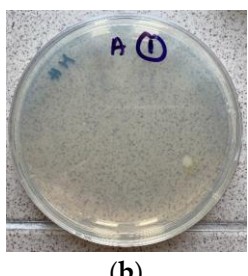 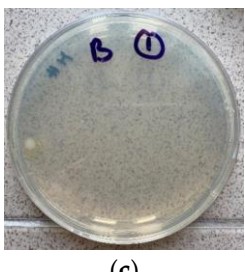 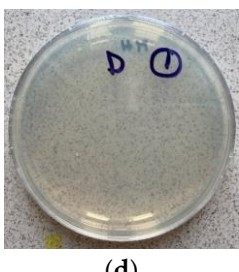 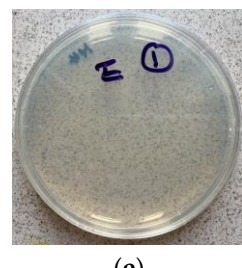

(a)     (b)     (c)     (d)     (e)

**Figure 4.** Microbial colonies after the disinfection with Zeta 7 Spray for each impression material: (**a**) Tropicalgin; (**b**) Zetaplus; (**c**) Oranwash L; (**d**) Elite HD + Putty Soft; (**e**) Elite HD + Light Body.

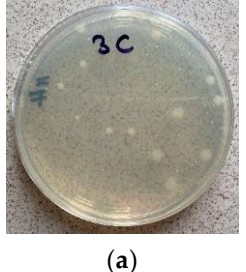
(a)

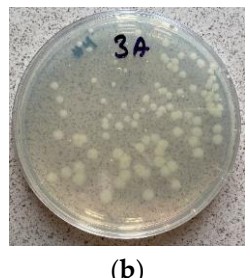
(b)

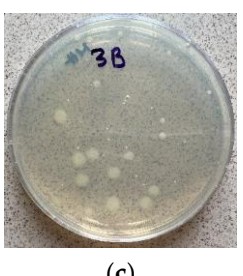
(c)

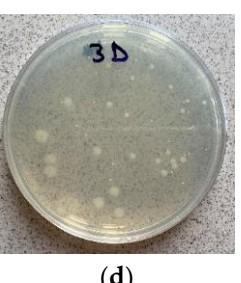
(d)

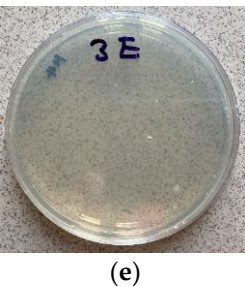
(e)

**Figure 5.** Microbial colonies after the disinfection with Zeta 3 Soft for each impression material: (**a**) Tropicalgin; (**b**) Zetaplus; (**c**) Oranwash L; (**d**) Elite HD + Putty Soft; (**e**) Elite HD + Light Body.

Table 5 and Figure 6 show the means of the colony numbers found on the Mueller–Hinton media for each disinfectant and impression material.

**Table 5.** Descriptive statistics of the values obtained after disinfection.

| Impression Materials | Disinfectants | Minimum | Median | Maximum | Mean | Std. Deviation | Lower 95% CI of Mean | Upper 95% CI of Mean |
|---|---|---|---|---|---|---|---|---|
| Tropicalgin | Zeta 7 Spray | 7 | 10 | 13 | 10 | 2 | 8.569 | 11.43 |
| | Zeta 7 Solution | 0 | 2 | 4 | 2 | 1.247 | 1.108 | 2.892 |
| | Zeta 3 Soft | 22 | 36 | 50 | 36 | 7.087 | 30.93 | 41.07 |
| Zetaplus | Zeta 7 Spray | 3 | 5 | 7 | 5 | 1.491 | 3.934 | 6.066 |
| | Zeta 7 Solution | 0 | 2 | 4 | 2 | 1.247 | 1.108 | 2.892 |
| | Zeta 3 Soft | 68 | 92.5 | 114 | 92 | 11.13 | 84.04 | 99.96 |
| Oranwash L | Zeta 7 Spray | 1 | 3 | 5 | 3 | 1.491 | 1.934 | 4.066 |
| | Zeta 7 Solution | 0 | 0 | 0 | 0 | 0 | 0 | 0 |
| | Zeta 3 Soft | 20 | 31 | 40 | 30 | 5.395 | 26.14 | 33.86 |
| Elite HD + Putty Soft | Zeta 7 Spray | 0 | 2.5 | 8 | 3 | 2.582 | 1.153 | 4.847 |
| | Zeta 7 Solution | 0 | 0.5 | 4 | 1 | 1.333 | 0.04619 | 1.954 |
| | Zeta 3 Soft | 30 | 49 | 68 | 49 | 14.5 | 38.63 | 59.37 |
| Elite HD + Light Body | Zeta 7 Spray | 0 | 1 | 3 | 1 | 1.054 | 0.2459 | 1.754 |
| | Zeta 7 Solution | 0 | 0 | 0 | 0 | 0 | 0 | 0 |
| | Zeta 3 Soft | 4 | 11.5 | 21 | 12 | 5.437 | 8.111 | 15.89 |

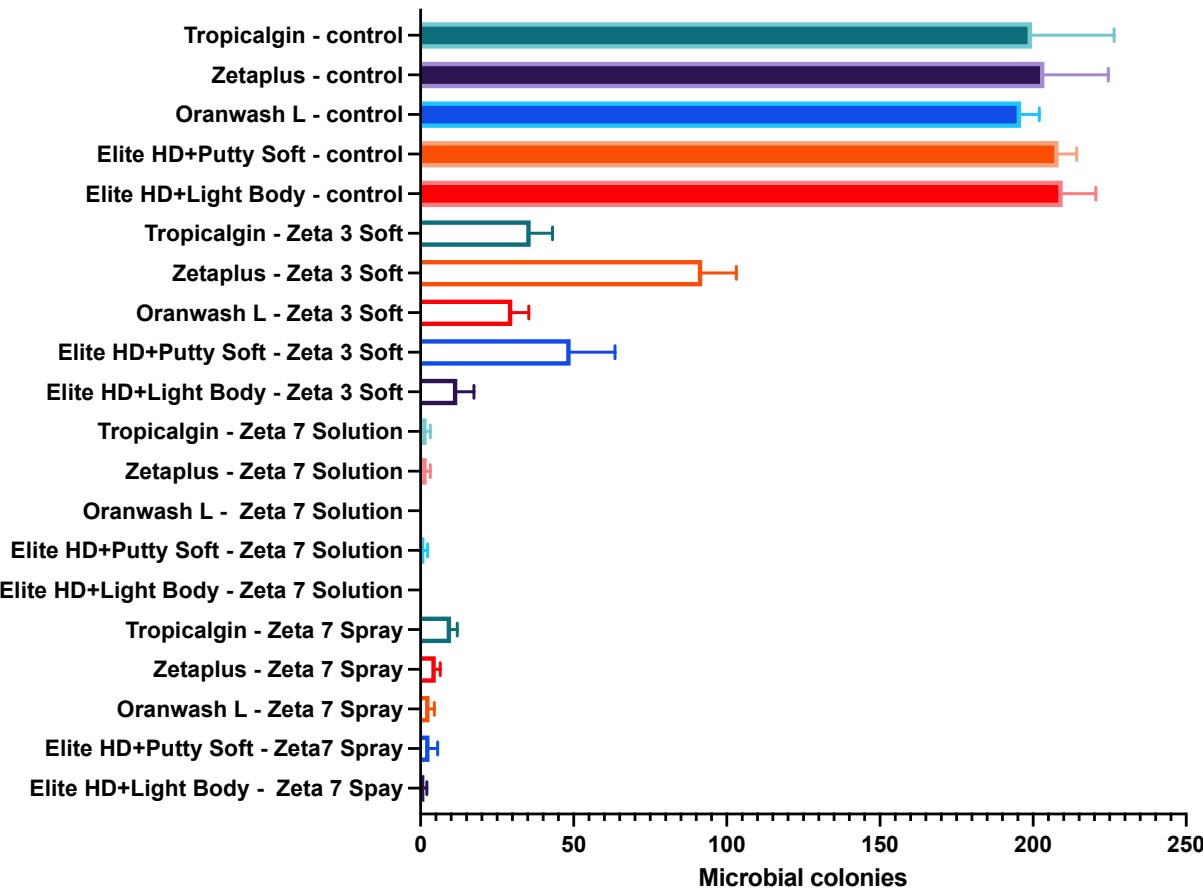

**Figure 6.** The number of microbial colonies for all impression materials before and after disinfection.

It was found that for all impression materials disinfected with Zeta 3 Soft the mean number of colonies is high, with no statistical difference with the values obtained after disinfection with the control groups. The $p$ values registered in this case after application of the Kruskal–Wallis test, followed by Dunn's multiple-comparisons test, are as follows: Tropicalgin–0.3326, Zetaplus–0.3311, Oranwash L–0.3216, Elite HD + Putty Soft–0.3255, Elite HD + Light Body–0.3019 (Figure 6).

A statistically significant difference was found between all values, using the Kruskal–Wallis test ($p < 0.001$). The results of Dunn's multiple-comparisons test are presented in Table 6.

**Table 6.** The comparison of the results of the disinfection efficacy between the three used disinfectants.

| | Zeta 7 Spray vs. Zeta 7 Solution | Zeta 7 Spray vs. Zeta 3 Soft | Zeta 7 Solution vs. Zeta 3 Soft |
|---|---|---|---|
| | | $p$ Value | |
| Tropicalgin | 0.0329 * | 0.0329 * | <0.0001 **** |
| Zetaplus | 0.0746 | 0.0206 * | <0.0001 * |
| Oranwash L | 0.0288 * | 0.0288 * | <0.0001 * |
| Elite HD + Putty Soft | 0.601 | 0.0041 ** | <0.0001 * |
| Elite HD + Light Body | 0.323 | 0.0039 ** | <0.0001 * |

* significant, ** very significant, **** extremely significant.

## 4. Discussion

The most common method of impression disinfection is the chemical method, due to its simple applicability. The chemical-agent efficacy is influenced mainly by the type and

the number of microorganisms, the concentration and the action time of the disinfectants, and the impression material for which it is used [15]. It is recommended that to label the impressions sent to the dental laboratory, dentists inform the dental technician about the disinfection status. This is to control the cross-infections, and avoid multiple disinfections, which can compromise the dimensional stability and the surface characteristics of the materials [16]. Several studies demonstrated that simply washing the silicone impression with water does not effectively reduce the microbial load of the material surface, and is not suitable for preventing cross-infections [17,18]. Our findings are in concordance with this research. For disinfection, alcohol, sodium hypochlorite, and glutaraldehyde-based disinfectants are more effective. The disinfection of alginate and polyvinyl-siloxane impressions with these disinfectants will result in adequate disinfection without compromising the quality of the materials [18,19]. Gupta et al. demonstrated that Indian dental technicians have insufficient knowledge about contamination possibilities through impressions [20]. Al Mortadi et al. recommend education programs for dental technicians concerning the importance of impression disinfection [6].

Our results demonstrate the efficacy of the two examined, specific disinfectants (Zeta 7 Spray and Zeta 7 Solution) in reducing the number of microbial colonies on the impression material's surface. According to our results, for two impression materials (Oranwash L and Tropicalgin), the Zeta 7 Solution is more appropriate for eliminating surface microorganisms. In the case of the other impression materials used, no significant differences were recorded between the action of the Zeta 7 solution and the Zeta 7 Spray. In a similar study to ours, conducted by Wezgowiec et al., the antimicrobial efficacy of UVC radiation, gaseous ozone, and liquid chemicals used to disinfect silicone dental-impression materials was evaluated. In this study, the C-silicone used was Oranwash L, Zetaplus, and A-silicone Panasil (putty, medium, light). A mixture of three microbial strains for sample inoculation, *P. aeruginosa* ATCC 27853, *S. aureus* ATCC 6538, and *C. albicans* ATCC 10231 (ATCC, Manassas, VA, USA) was used. The chemical solutions used for disinfection were Zeta 7 solution and Zeta 7 spray. The reported result was in accordance with ours, in that all disinfection methods evaluated were effective against selected oral pathogens (*P. aeruginosa, S. aureus, and C. albicans*) which were used to contaminate both types of C-silicone [21].

The Zeta 7 Solution, based on quaternary ammonium salts, reduced almost entirely the microorganisms load from the impressions surfaces; in the case of Oranwash L and Elite HD+ Light Body, no microorganisms were found on the culture media. According to Demajo et al. [22] and Samra and Bhide's research [23], immersion in disinfectant is one of the safest disinfection methods. Their literature states that the porosity of dental-impression materials leads to the penetration of microorganisms. By using the spraying method, the disinfectant cannot reach the entire surface of the impression. In most cases, the microorganisms penetrated through the porosities of the material into the deeper layers and will remain inside the impression material. On the other hand, Ulgey et al. demonstrated that using Zeta 7 Spray is the most effective disinfection method for the impression materials contaminated with *Pseudomonas aeruginosa* [24].

The tested solutions were effective in the case of alginate impressions. The immersion of the impressions resulted in a higher microbial reduction, as Giammanco et al. demonstrated in their study [25]; similar results were obtained in our study. The microorganisms on the surface of the dental impressions were reduced almost completely, when using the immersion-disinfection method. Ten minutes of immersion time did not affect the dimensional accuracy. After this period, the alginate impression must be immediately poured [26]. Some dentists avoid immersion because they believe that the impression will no longer have dimensional stability. According to Hussein et al., immersing an alginate impression in a disinfectant solution for 10 min does not affect the dimensional stability of the material [26].

More bacterial colonies were found on the surface of the condensation-silicone specimens, compared to the polyvinyl-siloxane specimens for each disinfectant used. This is

probably due to a higher porosity of the material, which allows a deeper penetration of the microorganisms, where the disinfectants cannot act with high efficacy.

A difference was also noticed between the number of colonies found on the surface of the light body compared with the putty consistency from the same type of silicone. This is probably due to the higher porosity of the putty materials, compared to the light-body ones. The infection control for polyvinyl siloxane materials can be done without considerable dimensional changes by autoclaving. The pouring of the casts must be delayed by 24 h to gain compensatory expansion [27]. Several studies demonstrated the dimensional stability of the addition silicones after a shorter immersion period than 60 min in 2% glutaraldehyde, sodium hypochlorite, or phenol solutions [27]. A chemical disinfection with 2% glutaraldehyde will result in more considerable dimensional changes than autoclaving [28]. Kavita et al. reported maximum stability after using different disinfectants for the heavy-body polyvinyl-siloxane impression materials [29]. Zeta 3 Soft spray is not adequate for impression disinfection; a high number of bacterial colonies were found on the surface of all impressions disinfected with this spray. Disinfecting the impressions with Zeta 3 Soft spray leads to a high risk of cross-contamination.

An in-vivo study conducted by Azevedo et al. showed that water wash reduced the microbial load by 11.7% (with no statistical difference). In comparison, all the other disinfectants used in the study reduced the microbial load by more than 99.9% [17]. Similar results were obtained in our in vitro study.

In in vivo research conducted by Jeyapalan et al., the results indicated that freshly prepared electrolyzed-oxidizing-water is a very effective method to reduce the microbial load (the microbial-reduction rate was 100%) in the case of addition-silicone impressions [30]. New ultraviolet disinfection-methods can be used to disinfect impressions without compromising their dimensional stability [23]. Gaseous-ozone treatment is also a promising method for disinfecting polyvinyl-siloxane impressions, which can increase the wettability of the material [31]. A useful disinfection method can be microwave irradiation, especially when combined with $H_2O_2$, which functions without adversely affecting the physical properties of dental-impression materials [32].

Disinfection with a nonthermal atmospheric-pressure plasma jet proved an effective disinfection tool for dental-impression material. With this method, the number of bacteria was significantly reduced, but the chemical properties of the surface of the impression material were changed without modification of the physical properties [33].

Limitations of the study:

- In vivo conditions could not be reproduced perfectly. The presence of saliva, blood, dental plaque, and gingival fluids were not taken into account.
- The smaller size and different form of the specimens, compared with an in vivo impression, can lead to different results, especially when the disinfection is carried out by spraying.
- The study was performed by examining the bacterial load of the impressions for three species of microorganisms: *Staphylococcus aureus*, *Pseudomonas aeruginosa*, and *Candida albicans*.

## 5. Conclusions

- Rinsing dental impressions with tap water leaves a considerable microbial load on the surface of the impression materials.
- The surface disinfectant tested is not effective for disinfecting dental impressions. These findings require further in-vivo examination to elucidate the disinfection efficacy in cases of other types of microorganisms from the oral cavity's bacterial flora and the disinfectants' effect on the dimensional stability of the impressions.

**Author Contributions:** Conceptualization, D.C. and O.P.; methodology, C.N.C. and D.C.; formal analysis, M.D.B.; investigation, C.N.C.; resources, K.M.J.; data curation, A.I.A.; writing—original draft preparation, O.P. and A.E.C.; writing—review and editing, K.M.J.; visualization, L.G.H.; supervision, D.C. All authors have read and agreed to the published version of the manuscript.

**Funding:** This research received no external funding.

**Institutional Review Board Statement:** Not applicable.

**Informed Consent Statement:** Not applicable.

**Data Availability Statement:** The dataset analyzed during the study is available from the corresponding author on request.

**Acknowledgments:** The authors declare no financial affiliation or involvement with any commercial organization with a direct financial interest in the materials discussed in this manuscript.

**Conflicts of Interest:** The authors declare no conflict of interest.

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
