# Peer review of "The Efficacy of Three Types of Disinfectants on the Microbial Flora from the Surface of Impression Materials Used in Dentistry—In Vitro Study"

_applsci, doi:10.3390/app13021097_

Round 1

Reviewer 2 Report

I believe that the presented in vitro research is a good and useful introduction to extending research to in vivo, especially taking into account the clinical significance. Despite the implementation of new technologies in dentistry, such as intraoral scanners, impressions using various masses are still often used. I recommend that the authors refer (apart from the limitation of the study) to the appropriateness of presenting similar in vivo studies taking into account different groups (such as patients without odontogenic foci vs. patients with periodontitis, etc.). I am for the publication of the article.

Reviewer 3 Report

Dear authors,

The disinfection of the impression materials, before sending the impression in the lab is always a critical and important step of the prosthodontics practice, therefore a study on the most efficient and proper method is always interesting. Testing a surface disinfectant was a clever addition, and adds to the originality of the topic. Indeed, as surface disinfection sprays may be often used as a handy means of disinfection and it is useful to know if their efficiency is adequate.

Please consider the following questions/comments:

How did you check or control the voids entrapped into the specimens after mixing the impression materials? Most of your materials were hand-mixed, and this may result in porosity within the specimens, even more so, as they are of a considerable thickness. The presence of voids could affect the microbial as well as the disinfectant penetration into the bulk of the specimen. More flat specimens would permit more control of the homogeneity of the material, as well as closer resemblance to the clinical conditions.

How did you quantify the amount of disinfection material that was sprayed over the specimens, in the case of spray disinfectants?

Because the impression materials were left to set at ambient temperature, it would be advisable to extend the recommended setting time by one or two minutes, to ensure that they were fully set.

Take a look at the paragraph staring at line #72. The description of the procedure involves one of the disinfectants, but this is what you did in all cases. The readers may find it confusing.

Your results are well presented and properly interpreted. I recommend that the statistically significant differences among the results to be presented more clearly, in the form of superscripts in the Tables, or as lines connecting the significant differences in the graph. Please check also again the differences between the control and Zeta 3 Soft. It seems to me, observing the means and standard deviations, that, at least some of them, could be significant at the 0.05 level.

The first conclusion seems somewhat arbitrary, since you did not check the level of contamination prior to rinsing with water, and thus you do not know if there was a reduction in the contamination level. Please rephrase, such as: ‘Rinsing dental impressions with tap water leaves a considerable microbial load on the surface of the impression materials’.

Limit the conclusions to the materials you have researched: ‘The surface disinfectant tested is not effective for disinfecting dental impressions’.
